# A Novel Endoscopic Ultrasonography Imaging Technique for Depicting Microcirculation in Pancreatobiliary Lesions without the Need for Contrast-Enhancement: A Prospective Exploratory Study

**DOI:** 10.3390/diagnostics11112018

**Published:** 2021-10-30

**Authors:** Yasunobu Yamashita, Takanori Yoshikawa, Hirofumi Yamazaki, Yuki Kawaji, Takashi Tamura, Keiichi Hatamaru, Masahiro Itonaga, Reiko Ashida, Yoshiyuki Ida, Takao Maekita, Mikitaka Iguchi, Masayuki Kitano

**Affiliations:** 1Second Department of Internal Medicine, Wakayama Medical University, Wakayama 641-8509, Japan; hirofumi.y.nagoya@gmail.com (H.Y.); y.kawaji1985@gmail.com (Y.K.); ttakashi28@gmail.com (T.T.); papepo51@wakayama-med.ac.jp (K.H.); itonaga@wakayama-med.ac.jp (M.I.); rashida@wakayama-med.ac.jp (R.A.); y-mori@wakayama-med.ac.jp (Y.I.); maekita@wakayama-med.ac.jp (T.M.); mikitaka@wakayama-med.ac.jp (M.I.); kitano@wakayama-med.ac.jp (M.K.); 2Clinical Study Support Center, Wakayama Medical University Hospital, Wakayama 641-8509, Japan; ta-yoshi@wakayama-med.ac.jp

**Keywords:** detective flow imaging endoscopic ultrasonography, doppler endoscopic ultrasonography, vessel detection, novel technique, contrast-enhanced endoscopic ultrasonography, pancreatobiliary lesion

## Abstract

Detective flow imaging endoscopic ultrasonography (DFI-EUS) provides a new method to image and detect fine vessels and low-velocity blood flow without using ultrasound contrast agents. The aim of this study was to evaluate the utility of DFI-EUS for pancreatobiliary lesions and lymph nodes. Between January 2019 and January 2020, 53 patients who underwent DFI-EUS, e-FLOW EUS, and contrast-enhanced EUS were enrolled. The ability of DFI-EUS and e-FLOW EUS to detect vessels was compared with that of contrast-enhanced EUS. This article describes the DFI technique along with our first experience of its use for vascular assessment of pancreatobiliary lesions. Vessels were imaged in 34 pancreatic solid lesions, eight intraductal papillary mucinous neoplasms (IPMNs), seven gall bladder lesions, and four swollen lymph nodes. DFI-EUS (91%) was significantly superior to e-FLOW EUS (53%) with respect to detection of vessels (*p* < 0.001) and for discrimination of mural nodules from mucous clots in IPMN and gallbladder lesions from sludge (*p* = 0.046). Thus, DFI-EUS has the potential to become an essential tool for diagnosis and vascular assessment of various diseases.

## 1. Introduction

Ultrasonography (US) is a common non-invasive and radiation-free technique for diagnosis. Endoscopic ultrasonography (EUS) was developed in the 1980s to overcome the problems caused by intervening gas, bone, and fat, which have adverse effects on transabdominal US images. EUS is thought to be one of the most reliable and efficient diagnostic modalities for pancreatobiliary disease because of its superiority to other modalities with respect to spatial resolution [1,2,3]. Therefore, EUS is useful for detecting small lesions. However, it is difficult to obtain a differential diagnosis based solely on EUS imaging characteristics, because most lesions appear hypoechoic on EUS.

Evaluation of vascularity is another approach to differential diagnosis. In this context, color-Doppler EUS (CD-EUS), power Doppler EUS, or e-FLOW EUS may be useful for observing vascularity in real time. The e-FLOW technique has the best resolution among these three modes; this is because it is a type of directional power Doppler US that provides better spatial and temporal resolution than conventional color and power Doppler US [4]. However, these conventional EUS Doppler modes are not good for visualizing fine vessels and slow flow. Contrast-enhanced EUS (CE-EUS) increases the detectability of vessels and can overcome some of these limitations. Indeed, many studies demonstrate the utility of CE-EUS for diagnosis of pancreatobiliary lesions [5,6,7,8,9,10,11,12,13,14,15,16,17,18,19,20,21,22,23]; thus, the technique is used widely in the clinic. However, there are some problems. For example, the use of contrast agent brought to 19 cases of severe adverse events (0.012%) and three cases of fatal adverse events (0.002%) in 157,838 cases [24]; the contrast agent used is relatively expensive (approximately $150 per procedure); special equipment is necessary; and the procedure requires more time than routine EUS due to the intravenous injection of the contrast agent. In response to these issues, detective flow imaging EUS (DFI-EUS) has been developed, and a recent study demonstrates its utility for vascular assessment of submucosal tumors [25]. However, no studies have examined the utility of DFI-EUS for pancreatobiliary lesions. Here, we examined the utility of DFI-EUS for pancreatobiliary lesions and lymph node swelling and compared the results with those of e-FLOW, which is another kind of directional power Doppler ultrasonography with greater spatial and temporal resolution than conventional color and power Doppler EUS) [4] and CE-EUS.

## 2. Materials and Methods

### 2.1. Patients

Fifty-three patients (23 men and 30 women) who underwent DFI-EUS, e-FLOW-EUS, and contrast-enhanced EUS at Wakayama Medical University Hospital between January 2019 and January 2020 were enrolled prospectively. Vessel flow was assessed using all three modalities. The inclusion criteria were as follows: age > 20 years; and DFI-EUS and e-FLOW were performed on the same day. A final diagnosis of a malignant lesion was confirmed by histopathology; probable benign lesions were followed up for at least 12 months.

### 2.2. Study Design

This was a prospective study designed to evaluate the usefulness of DFI-EUS for imaging of pancreatobiliary lesions and lymph node swelling. The sensitivity of vessel detection by DFI-EUS was compared with that of e-FLOW EUS. Vascularity was confirmed by CE-EUS.

### 2.3. EUS Procedure

Ultrasound-equipped (ARIETTA 850; FUJIFILM Medical Co., Ltd., Tokyo, Japan) convex-type endoscopes (GF-UCT260; Olympus, Tokyo, Japan) were employed. Patients received EUS in the left lateral position under diazepam-induced sedation. Heart rate, blood pressure, and oxygen saturation were monitored throughout. Fundamental B-mode EUS was initially performed. After using the fundamental B-mode, we performed e-FLOW EUS and DFI-EUS. For DFI-EUS, dynamic range, transmission frequency, and color gain were adjusted to 83, 5, and 65 MHz, respectively. For e-FLOW EUS, these parameters were 83, 5.0, and 52 MHz, respectively. Optimal color gains were fixed to allow comparisons between the two modes under the same condition. However, random noise appeared in the lesion when we used the same color gains as those of DFI-EUS in e-FLOW EUS. Therefore, the optimal color gain was different between DFI-EUS and e-FLOW EUS. Before the study, the color gains necessary to eliminate random noise in the lesions were determined to be 65 and 52 MHz for DFI-EUS and e-FLOW EUS, respectively. The optimal color gain values for DFI-EUS and e-FLOW EUS were kept the same in all patients. Finally, CE-EUS with second generation contrast agent (Sonazoid^®^; GE healthcare, Tokyo, Japan) was performed with the extended pure harmonic detection method for which the mechanical index was set at 0.35. The endosonographer recorded all EUS images for later evaluation of vascularity.

### 2.4. DFI-EUS

DFI technology is a new innovative Doppler US technique. DFI technology detects microvessels with very slow flow states; it does this by using an algorithm that permits minute vessels to be visualized at slow velocity in the absence of motion artifacts and without contrast agents. DFI is based on the following principles. The signals of ultrasonic Doppler are derived not only from blood flow but also from motion artifacts. Motion artifact signals overlap with those of low-speed flow components. Overlap due to motion artifact, which are unnecessary signals, hinders the visualization of low-flow signals. (Figure 1). When conventional Doppler techniques remove motion artifacts with a single-dimensional wall filter, not only overlaying tissue motion artifacts but also the low-flow component are removed. Consequently, the visibility of flow in smaller vessels is lost due to loss of low-flow data. On the other hand, DFI can separate flow signals from motion artifacts with a multi-dimensional filter, which analyzes motion artifacts and uses an adaptive algorithm to identify and remove tissue motion. Therefore, DFI overcomes this loss of low-flow signals by separating them from overlapping tissue motion artifacts without jeopardizing the visualization of low-flow components and, in addition, provides detailed vessel signals (Figure 2). Conventional Doppler ultrasound techniques were designed principally to visualize blood flow at high resolution. As a result, e-FLOW is currently the method that provides the best resolution (Figure 3). Moving beyond this goal, DFI is also able to visualize lower-velocity blood flow (Figure 3). DFI reveals a more accurate depiction of blood flow in comparison with conventional Doppler image. This results in a high-resolution ultrasonography image in which minute vessels and low velocity flow can be seen clearly. All of this is done at high frame rates, which is impossible using other Doppler technologies. Thus, DFI expands the range over which blood flow can be visualized and is especially effective in visualizing low microvascular flow. Moreover, it gives clinicians a new way to visualize minute vessels when evaluating lesions and tumors.

### 2.5. Statistical Analysis

Statistical analysis was performed using JMP Pro version 13 (SAS Institute Inc., Cary, NC, USA). The Mann–Whitney U test was used to compare continuous variables between the two groups. Differences were considered significant when the *p* value was <0.05. The McNemar test was used to compare vessel detection between DFI-EUS and e-FLOW EUS.

## 3. Results

The final diagnoses of the 53 examined lesions were as follows: pancreatic cancer (*n* = 23), inflammatory mass in the pancreas (*n* = 8), solid-pseudopapillary neoplasm (*n* = 2), neuroendocrine tumor (*n* = 1), intraductal papillary mucinous neoplasm (IPMN) (*n* = 8), gall bladder cancer (*n* = 2), cholesterol polyp (*n* = 1), chronic cholecystitis (*n* = 1), gall bladder sludge (*n* =3), benign lymph node (*n* = 3), and malignant lymph node (*n* = 1) (Table 1).

The gallbladder sludge and the mucous clot in IPMN were diagnosed by CE-EUS in the absence of vascularity. Neither DFI-EUS nor e-FLOW EUS detected vascularity in these lesions. By contrast, the mural nodule in IPMN and gallbladder mass were diagnosed by CE-EUS when the lesion showed vascularity. DFI-EUS discriminated mural nodules (*n* = 2) from mucous clots (*n* = 3) in IPMN, and the gallbladder mass (*n* = 3; one cholesterol polyp and two cancers) from sludge (*n* = 3); all mural nodules in IPMN and gallbladder masses showed vascularity (100%) (Table 2). For IPMN and gallbladder lesions, DFI-EUS showed a more detailed image of the vessels than e-FLOW EUS in pacreatobiliary lesions (Figure 4, Figure 5, Figure 6 and Figure 7). By contrast, e-FLOW EUS detected vessels in only one of five in both mural nodules in IPMN and gallbladder masses (20%) (Table 2). For differential diagnoses between mural nodule and mucous clot in IPMN, and gallbladder masses and sludge, DFI-EUS was significantly superior to e-FLOW (*p* = 0.046). In addition, DFI-EUS provided more detailed detection/imaging of vessels in the normal pancreas than e-FLOW EUS (Figure 8).

For imaging of pancreatobiliary lesions (all lesions), the sensitivity, specificity, and accuracy for vessel detection by DFI-EUS (compared with CE-EUS) were 91%, 100%, and 92%, respectively. By contrast, the sensitivity, specificity, and accuracy for vessel detection by e-FLOW EUS (compared with CE-EUS) were 53%, 100%, and 60%, respectively. Compared with CE-EUS, DFI-EUS was significantly superior to e-FLOW EUS with respect to vessel detection in pancreatobiliary lesions (*p* < 0.001) (Table 2). The presence of vessels on DFI-EUS correlated significantly with that on CE-EUS (*p* = 0.025) (Table 3). By contrast, there were no significant correlation between e-FLOW EUS and CE-EUS (Table 4).

For pancreatic lesions, the sensitivity, specificity, and accuracy of DFI-EUS compared with CE-EUS for vessel detection were 89%, 100%, and 90%, respectively. By contrast, the sensitivity, specificity, and accuracy of e-FLOW EUS compared with CE-EUS were 51%, 100%, and 57%, respectively (Table 2). Compared with CE-EUS, DFI-EUS was significantly superior to e-FLOW for vessel detection in pancreatic lesions (*p* < 0.001). The presence of vessels on DFI-EUS correlated significantly with that on CE-EUS (*p* = 0.025). However, there were no significant correlations between e-FLOW EUS and CE-EUS.

## 4. Discussion

We examined the ability of DFI-EUS to detect vessels in 53 lesions. In all 53 cases, the vessel detection (when compared with CE-EUS) of DFI-EUS (91% in sensitivity, 100% in specificity, and 92% in accuracy, respectively) was superior to that of eFLOW EUS (53% in sensitivity, 100% in specificity, and 60% in accuracy, respectively). Thus, this study shows that, compared with e-FLOW EUS, DFI-EUS is significantly more effective for vessel detection; this is likely because DFI-EUS visualizes fine vessels much more clearly than conventional Doppler EUS (Figure 4, Figure 5, Figure 6, Figure 7 and Figure 8). However, DFI-EUS failed to detect vessels in four of 46 lesions (9%) in which vessels were detected by CE-EUS. In particular, the vessel detection rate was low on pancreatic cancer. This may be due to following reasons. Pancreatic cancer is a hypovascular tumor with more necrotic tissue than other tumors. Therefore, it may be difficult to detect by DFI-EUS because this type of tumor contains fewer vessels that other tumors. Moreover, the presence of a capsule, and heterogeneity in the internal echo of pancreatic cancer, may affect vessel detection in DFI-EUS. Therefore, a little more development time may be needed if DFI-EUS is to replace CE-EUS completely.

When assessing indications for surgical intervention for IPMN and gallbladder lesions, it is important to discriminate mural nodules from mucus clots in IPMN and gallbladder lesions from gallbladder sludge. Therefore, assessment of microvessels can provide valuable information that discriminates mural nodules from mucous clots in IPMN and gallbladder lesions from gallbladder sludge. Several studies report the utility of CE-EUS for differential diagnoses based on vascular assessment in pancreatobiliary diseases [11,12,13,14,15,16,17,18]. Here, we found that DFI-EUS (100%) was significantly superior to e-FLOW EUS (64%) for differential diagnosis with detecting vascularity. As such, DFI-EUS may be a useful alternative to CE-EUS for evaluating microvascularity of lesions for differential diagnosis. CE-EUS was reported to be useful for obtaining a differential diagnosis between an iso-vascular inflammatory mass and a hypo-vascular pancreatic tumor in previous reports [5,6,9]. Therefore, quantitative analysis of vessels in tumors with DFI-EUS may be useful for differential diagnosis of pancreatic tumors.

In Doppler imaging, visualization of vessels is difficult if the color gain is below a certain threshold. Conversely, the higher the vessel signal display, the stronger the random noise, which prevents the visualization of vessels with a slower flow [26]. In the present study, we determined the optimal settings of color gain that prevented random noise appearing in lesions using either mode. Additionally, the optimal color gain was kept constant in all cases to prevent the effects of different color gains. However, within a certain color gain range, the sensitivity with which vessels can be visualized is not ostensibly related to color gain but to how well vessel signals can be displayed in the same vessels [26]. Changes in color gain may not affect the sensitivity of vessel visualization.

Previous reports have compared the utility of transabdominal US for microvascular imaging with that of conventional Doppler imaging for the diagnosis of several diseases [27,28,29,30,31,32]. Another microvascular imaging technique similar to DFI, super microvascular imaging, furnishes significantly more information about vascularity than conventional Doppler imaging, making it more suitable for the differential diagnosis of lesions of the breast, thyroid, liver, kidney, and cervical lymph nodes [27,28,29,30,31]. However, performing transabdominal US to assess microvessels is difficult if the patient is obese (due to the depth of subcutaneous fat). The increased distance between the deeper organs and the probe also has a negative effect on the clarity of the microvascular images [32]. Therefore, to date, no studies into the utility of transabdominal US for microvascular imaging of pancreatobiliary lesions have been reported. EUS circumvents the problems of transabdominal US imaging, which are due mainly to the extensive distance between the probe and pancreatobiliary lesion, as well as intervening bone, fat, and gas. On the other hand, only DFI is available on EUS among several microvascular imaging techniques. These points make DFI-EUS more suitable for assessment of microvascular in pancreatobiliary lesions. However, there is only one study for submucosal tumor due to the new technique and due to new imaging technology. [25].

This study has several limitations. First, it was a single-center study that had a small sample size for various pancreatobiliary lesions and lymph node swellings. Second, optimal color gains were fixed to allow comparisons between the two modes under the same condition. However, the optimal color gains of both modes may not have been identical for each case. Because this report is the first to describe the use of DFI-EUS for imaging of pancreatobiliary lesions, our data can be considered as exploratory. Wider application of the technique, along with further largescale investigations, are needed to standardize the methods used for image acquisition and interpretation and to achieve consensus regarding clinically practical diagnostic criteria.

## 5. Conclusions

DFI-EUS is an extremely useful tool that provides valuable information about fine- and slow-flow vessels that cannot be evaluated using conventional Doppler-mode EUS. In the near future, DFI-EUS will be an essential tool for diagnosis and assessment of various diseases and a useful alternative to CE-EUS for vascular imaging.

## Figures and Tables

**Figure 1 diagnostics-11-02018-f001:**
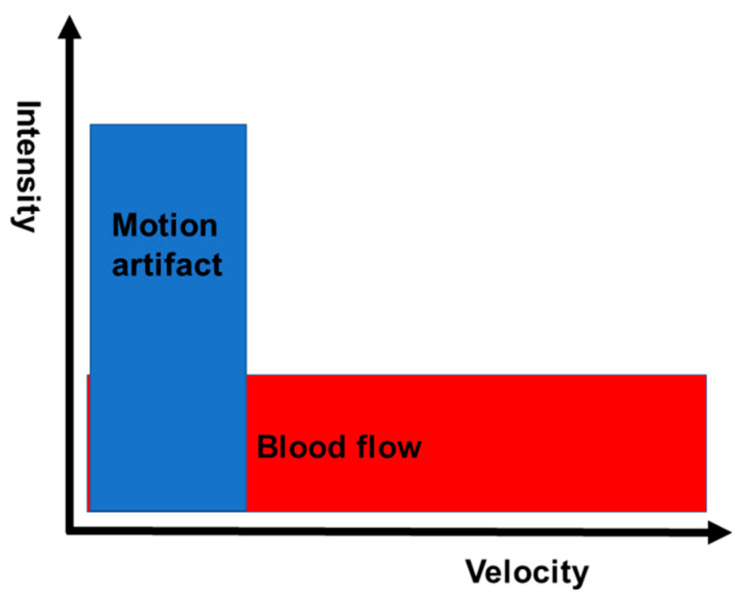
Doppler signals. The signals from motion artifacts overlap the low-speed flow components. Overlap due to motion artifact, which are unnecessary signals, hinders the visualization of low-flow signals.

**Figure 2 diagnostics-11-02018-f002:**
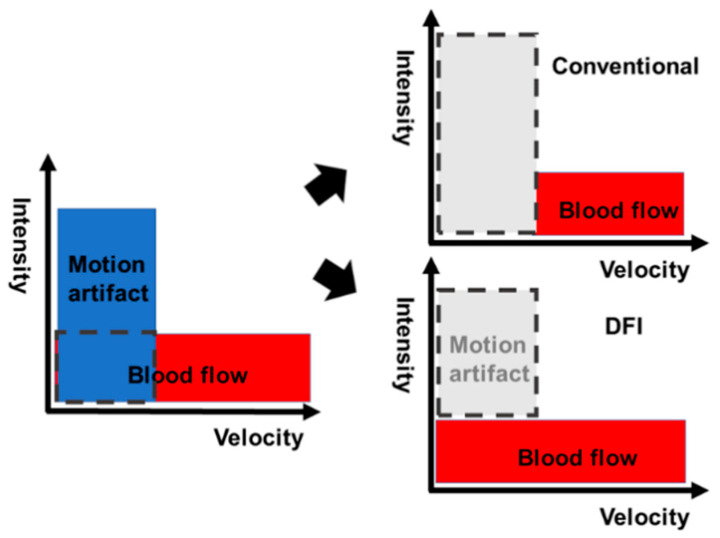
Conventional Doppler and DFI images. When conventional Doppler techniques remove motion artifacts with a single-dimensional wall filter, not only overlaying tissue motion artifacts but also the low-flow component are removed. Consequently, the visibility of flow in smaller vessels is lost due to loss of low-flow data. On the other hand, DFI can separate flow signals from motion artifacts with a multi-dimensional filter, which analyzes motion artifacts and uses an adaptive algorithm to identify and remove tissue motion. Therefore, DFI overcomes this loss of low-flow signals by separating them from overlapping tissue motion artifacts without jeopardizing the visualization of low-flow components and, in addition, provides detailed vessel signals.

**Figure 3 diagnostics-11-02018-f003:**
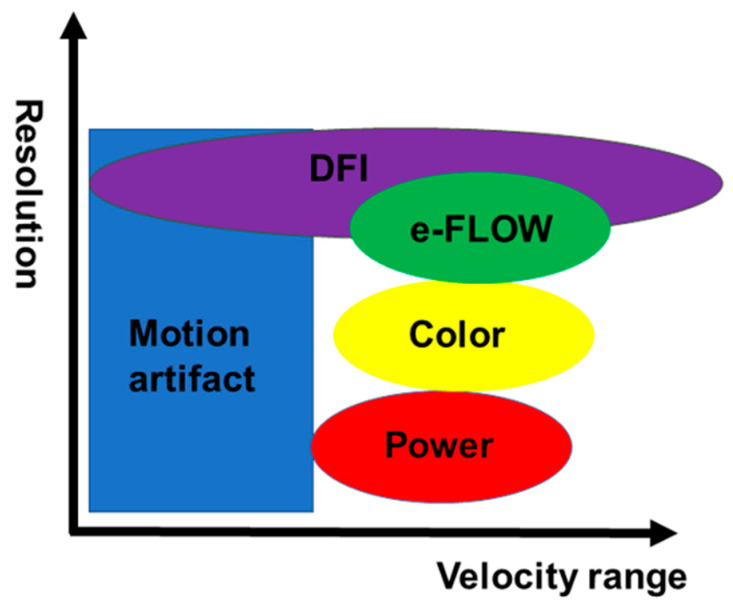
Comparison of DFI with conventional techniques. Conventional Doppler techniques are developed with the primary goal of visualizing blood flows at higher resolution. As a result, e-FLOW is currently the method that provides the best resolution. Moving beyond this goal, DFI is also able to visualize lower-velocity blood flow. DFI reveals a more accurate depiction of blood flow in comparison with conventional Doppler image.

**Figure 4 diagnostics-11-02018-f004:**
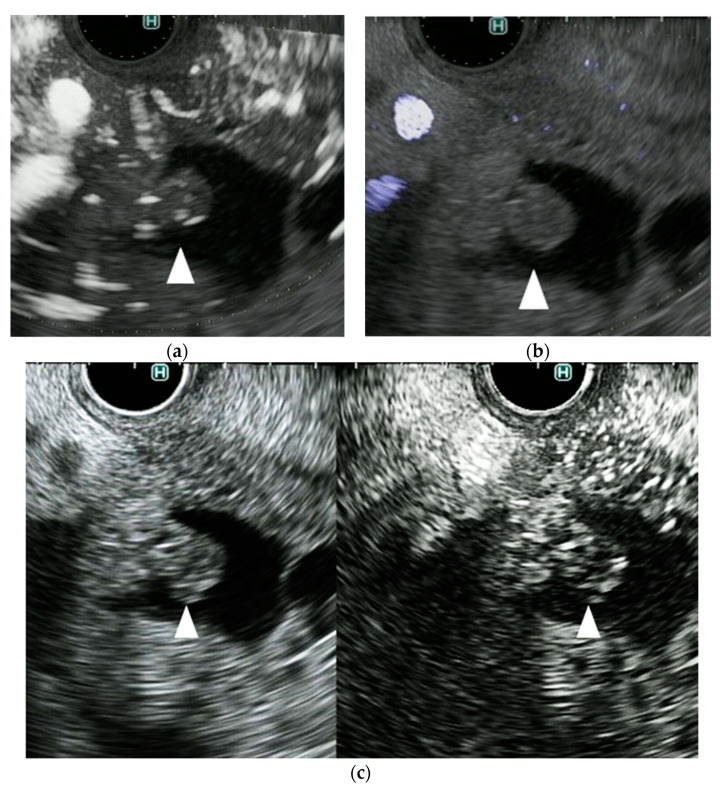
A representative case of a mural nodule in a patient with an intraductal papillary mucinous neoplasm (IPMN). (**a**) DFI endoscopic ultrasonography shows vascularity in the mural lesion (arrow). (**b**) e-FLOW endoscopic ultrasonography shows no vascularity in the mural lesion (arrow). (**c**) Contrast-enhanced endoscopic ultrasonography shows vascularity in the mural lesion (arrow).

**Figure 5 diagnostics-11-02018-f005:**
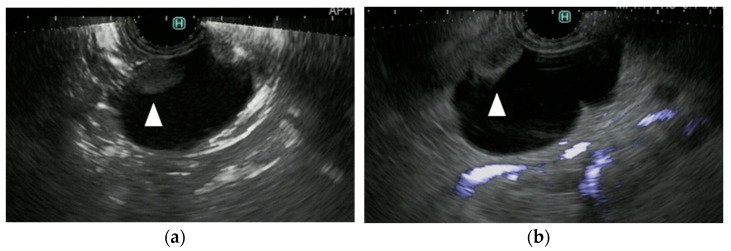
A representative case of a mucous clot in a patient with an intraductal papillary mucinous neoplasm (IPMN). (**a**) DFI endoscopic ultrasonography shows no vascularity in the mural lesion (arrow). (**b**) e-FLOW endoscopic ultrasonography shows no vascularity in the mural lesion (arrow). (**c**) Contrast-enhanced endoscopic ultrasonography shows no vascularity in the mural lesion (arrow).

**Figure 6 diagnostics-11-02018-f006:**
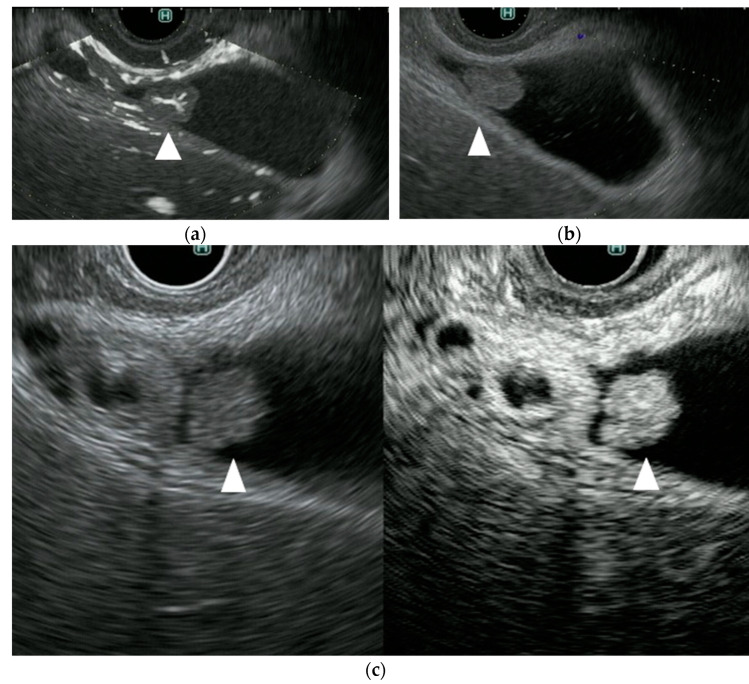
A representative case of a gallbladder cancer. (**a**) DFI endoscopic ultrasonography shows vascularity in the gallbladder lesion (arrow). (**b**) e-FLOW endoscopic ultrasonography shows no vascularity in the gallbladder lesion (arrow). (**c**) Contrast-enhanced endoscopic ultrasonography shows vascularity in the gallbladder lesion (arrow).

**Figure 7 diagnostics-11-02018-f007:**
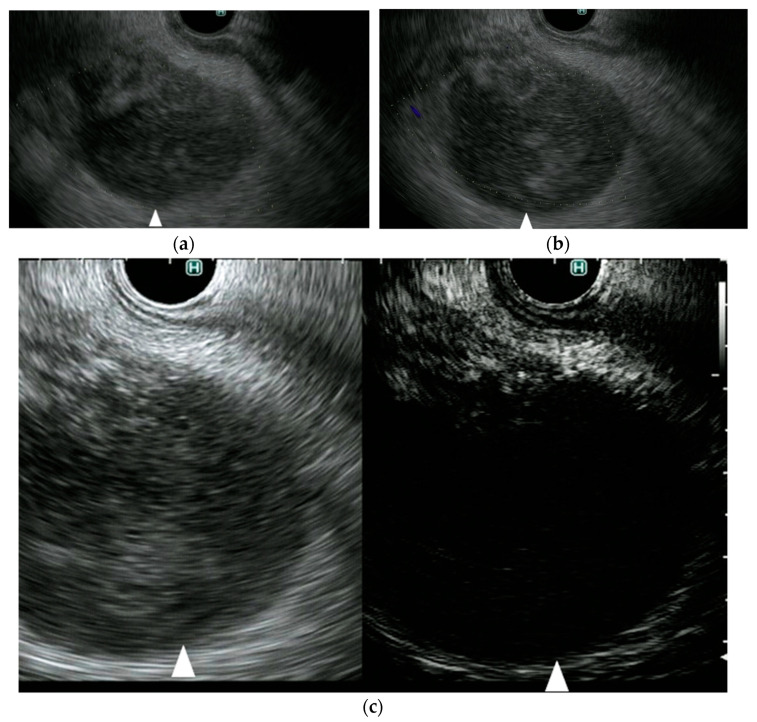
A representative case of gallbladder sludge. (**a**) DFI endoscopic ultrasonography shows no vascularity in the gallbladder lesion (arrow). (**b**) e-FLOW endoscopic ultrasonography shows no vascularity in the gallbladder lesion (arrow). (**c**) Contrast-enhanced endoscopic ultrasonography shows no vascularity in the gallbladder lesion (arrow).

**Figure 8 diagnostics-11-02018-f008:**
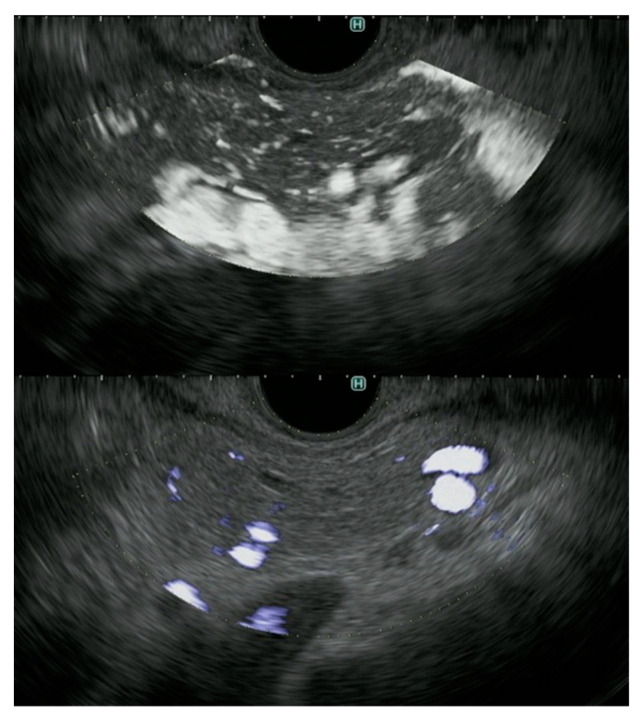
Detection of vessels in normal pancreas by DFI and e-FLOW endoscopic ultrasonography. DFI endoscopic ultrasonography (**upper**) provides better detection of vessels in normal pancreas than e-FLOW endoscopic ultrasonography (**lower**).

**Table 1 diagnostics-11-02018-t001:** Patient characteristics.

Characteristic	
Median age (range), years	69 (20–90)
Sex (male:female)	23/30
Diagnosis	
Pancreatic cancer	23
Inflammatory mass	8
Solid papillary neoplasm	2
Neuroendocrine tumor	1
Intraductal papillary mucinous neoplasm	8
Gallbladder cancer	2
Cholesterol polyp	3
Chronic cholecystitis	1
Gallbladder sludge	3
Benign lymph node	4

**Table 2 diagnostics-11-02018-t002:** Vascular assessment of pancreatobiliary lesions by DFI-EUS, e-FLOW EUS, and CE-EUS.

	DFI-EUS	e-FLOW EUS	CE-EUS
Pancreatic cancer	83% (19/23)	52% (12/23)	100% (23/23)
Neuroendocrine tumor	100% (1/1)	100% (1/1)	100% (1/1)
Solid papillary neoplasm	100% (2/2)	50% (1/2)	100% (2/2)
IPMN with mural nodule	100% (2/2)	0% (0/2)	100% (2/2)
IPMC	100% (1/1)	0% (0/1)	100% (1/1)
IPMN with mucous clot	0% (0/5)	0% (0/5)	0% (0/5)
Inflammatory mass	100% (8/8)	63% (5/8)	100% (8/8)
Malignant lymph node	100% (1/1)	100% (1/1)	100% (1/1)
Benign lymph node	100% (3/3)	67% (2/3)	100% (3/3)
Gallbladder cancer	100% (2/2)	50% (1/2)	100% (2/2)
Gallbladder cholesterol polyp	100% (1/1)	0% (0/1)	100% (1/1)
Chronic cholecystitis	100% (1/1)	100% (1/1)	100% (1/1)
Gallbladder sludge	0% (0/3)	0% (0/3)	0% (0/3)

IPMN, intraductal papillary mucinous neoplasm; IPMC, intraductal papillary mucinous carcinoma; DFI-EUS, detective flow imaging endoscopic ultrasonography; CE-EUS, contrast-enhanced EUS. DFI-EUS: sensitivity, 91%; specificity, 100%: accuracy, 92%. e-FLOW EUS: sensitivity, 53%; specificity, 100%: accuracy, 60%. *p* < 0.001 (DFI-EUS vs. e-FLOW EUS, compared with CE-EUS).

**Table 3 diagnostics-11-02018-t003:** Correlation between visualization of vessels in pancreatobiliary lesions between DFI-EUS and CE-EUS.

	Vessels Positive on CE-EUS	Vessels Negative on CE-EUS
Vessels positive on DFI-EUS	41	0
Vessels negative on DFI-EUS	4	8

*p* = 0.025. DFI-EUS, detective flow imaging endoscopic ultrasonography; CE-EUS, contrast-enhanced EUS.

**Table 4 diagnostics-11-02018-t004:** Correlation between visualization of vessels in pancreatobiliary lesions by e-FLOW EUS and CE-EUS.

	Vessels Positive on CE-EUS	Vessels Negative on CE-EUS
Vessels positive on e-FLOW EUS	24	0
Vessels negative on e-FLOW EUS	21	8

CE-EUS, contrast-enhanced EUS.

## Data Availability

No new data were created or analyzed in this study. Data sharing is not applicable to this article.

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
