# Peer review of "A Novel Endoscopic Ultrasonography Imaging Technique for Depicting Microcirculation in Pancreatobiliary Lesions without the Need for Contrast-Enhancement: A Prospective Exploratory Study"

_diagnostics, 2021, doi:10.3390/diagnostics11112018_

Round 1

Reviewer 1 Report

The authors try to evaluate a technique, named detective flow imaging endoscopic ultrasonography (DFI-EUS), for pancreatobiliary lesions and lymph nodes.

It is an interesting research.

However, the technical background is weak. For example, figures 1-3 are descriptive.

How to separate tissue motion and low-flow components?

The technique may detect the blood flow in small vessels. How that relates assessment of pancreatobiliary lesions?

Author Response

1." However, the technical background is weak. For example, figures 1-3 are descriptive."

Reply: Thank you for your comment. We have added detailed information regarding DFI and in the legends of Figures 1–3 (Page 3, Lines 102–121).

2." How to separate tissue motion and low-flow components?"

Reply: Thank you for your informative comment. DFI can separate flow signals from motion artifacts with a multi-dimensional filter, which analyzes motion artifacts and uses an adaptive algorithm to identify and remove tissue motion (Page 3, Lines 110–112).

3." The technique may detect the blood flow in small vessels. How that relates assessment of pancreatobiliary lesions "

Reply: Thank you for your important comment. Several studies reported the utility of contrast-enhanced EUS for differential diagnoses based on vascular assessment in pancreatobiliary diseases (Page 10, Lines 262–264). DFI can visualize microcirculation without contrast enhancement. The presence or absence of mural nodules is an important factor for making decisions regarding surgical intervention for IPMNs in international consensus guidelines. However, it is difficult to discriminate mural nodules from mucous clots with B mode EUS. Contrast-enhanced EUS was superior to conventional Doppler EUS and useful for determining surgical indications when the presence or absence of vascularity in mural lesions was assessed using these two techniques (Reference No. 13). Our study indicated that DFI (100%) is useful for vessel detection in IPMN mural nodules compared with eFLOW (0%) (Table 2). Therefore, DFI is expected to be a useful tool for differential diagnosis between mural nodules and mucous clots in IPMNs. Moreover, DFI is expected to be a new diagnostic tool for pancreatobiliary diseases where the usefulness of contrast-enhanced EUS has been reported. However, it is difficult to make conclusions about the utility of DFI in this study because this was an exploratory study with a small sample size. Therefore, further studies with larger cohorts from multiple centers are required.

Reviewer 2 Report

The paper is interesting and innovative, useful in daily clinical practice.
It only needs some corrections as suggested in the attached PDF

About the contents and therefore about minimal corrections it would be useful if the Authors clarified some aspects:

why color gains have a higher frequency for DFI-EUS? May be they could add it in paragraph 2.4

The second item concerns how the authors explain the low predictive value on pancreatic tumors

The third question concerns the role of DFI-EUS in discriminating tumors from inflammatory masses in pancreatitis.

These 2 topics should be added in the discussion

Author Response

Major

  1. " why color gains have a higher frequency for DFI-EUS?"

Reply: Thank you for this instructive comment. We determined the optimal color gain settings that prevented random noise appearing in lesions using either mode. Optimal color gains were fixed to allow comparisons between the two modes under the same condition. However, random noise appeared in the lesion when we used the same color gains as those of DFI-EUS in e-FLOW EUS. Therefore, the optimal color gain was different between DF-EUS and e-FLOW EUS (Page 2, Lines 85–88). However, within a certain color gain range, the sensitivity with which vessels can be visualized is not ostensibly related to color gain, but to how well vessel signals can be displayed in the same vessels. Changes in color gain may not affect the sensitivity of vessel visualization (Page 10, Lines 272–275).

2." The second item concerns how the authors explain the low predictive value on pancreatic tumors"

Reply: Thank you for this instructive comment. Pancreatic cancer is a hypovascular tumor with more necrotic tissue than other tumors. Therefore, it may be difficult to detect by DFI because this type of tumor contains fewer vessels that other tumors. Moreover, the presence of a capsule, and heterogeneity in the internal echo of pancreatic cancer may affect vessel detection in DFI (Page 7, Lines 214–219).

3." The third question concerns the role of DFI-EUS in discriminating tumors from inflammatory masses in pancreatitis."

Reply: Thank you for this important comment. Because this report is the first to describe the use of DFI-EUS for the imaging of pancreatobiliary lesions, our data can be considered as exploratory. The sensitivity of vessel detection by DFI-EUS was compared with that of e-FLOW, which showed that DFI-EUS was more superior to e-FLOW EUS than CE-EUS. In many previous studies, vascular assessment using CE-EUS was reported to be useful for obtaining a differential diagnosis between an iso-vascular inflammatory mass and a hypo-vascular pancreatic tumor. Therefore, quantitative analysis of vessels in tumors with DFI-EUS may be useful for differential diagnosis (Page 10, Lines 267–270). We plan to consider using DFI-EUS when the number of pancreatic tumors including inflammatory masses increases in the near future. 

Minor

  1. " Line 53: “contras” should be “contrast”.

Reply: We apologize for this mistake and have changed “contras” to “contrast”.

  1. Line 89: “agnent” should be “agent”.

Reply: We apologize for this mistake and have changed “agnent” to “agent”.

  1. Table 4: “Vessels positive won e-FLOW EUS” should be “Vessels positive on e-FLOW EUS”.

Reply: We apologize for this mistake and have changed “won” to “on”.

  1. Line 192: “Figure4-8” should be “Figures 4-8”."

Reply: We apologize for this mistake and have changed “Figure4-8” to “Figures 4-8”.

Round 2

Reviewer 1 Report

The authors have made some improvements.